# Patient–Nurse Communication in an Oncology Hospital Setting: A Qualitative Study

**DOI:** 10.3390/healthcare13010050

**Published:** 2024-12-30

**Authors:** Laura Iacorossi, Giovanna D’Antonio, Maria Condoleo, Lara Guariglia, Fabrizio Petrone, Simona Molinaro, Anita Caruso

**Affiliations:** 1Department of Life, Health and Health Professions Sciences, Link Campus University, 00165 Rome, Italy; l.iacorossi@unilink.it; 2Psychology Unit, IRCCS “Regina Elena” National Cancer Institute, 00144 Rome, Italy; giovanna.dantonio@ifo.it (G.D.); maria.condoleo@ifo.it (M.C.); anita.caruso@ifo.it (A.C.); 3Nursing, Technical, Rehabilitation, Assistance and Research Direction, IRCCS “Regina Elena” National Cancer Institute, 00144 Rome, Italy; fabrizio.petrone@ifo.it (F.P.); simona.molinaro@ifo.it (S.M.)

**Keywords:** communication, patient, nurse, oncology, competence, hospital setting

## Abstract

Background: Communication is an important aspect in making patients competent to define, process, and manage their disease condition as well as to intercept and satisfy psychosocial needs. Communication between patient and nurse is central to the learning and orientating process since the nurse has the greatest frequency and continuity of relationship with patients and their families. This study aims to investigate the quality of communication between patient and nurse and the factors that promote or hinder effective communication from the oncology patient’s perspective within an inpatient hospital setting. Methods: A descriptive qualitative study was conducted with one-to-one semi-structured interviews analyzed using the Framework Analysis methodology. The population consisted of oncologic patients admitted to the Medical Oncology Units of the Regina Elena National Cancer Institute in Rome. Data were analyzed using Ritchie and Spencer’s Framework Analysis. Results: The sample comprised 20 patients, with an average age of 61.35, admitted to the Medical Oncology Units of the Regina Elena National Cancer Institute in Rome. Three themes emerged: positive communication as an element of care, factors fostering communication, and factors hindering communication. Conclusions: The sample interviewed deems the quality of communication satisfactory. Familiar communication style, direct language, and caring are factors fostering communication. In contrast, lack of communication between medical and nursing staff, shortage of staff, and lack of time are considered communication barriers. Advanced communication competencies in nurses are crucial for effectively addressing the emotional and psychosocial needs of cancer patients, fostering a more empathetic and supportive care environment.

## 1. Introduction

The healthcare system reorganization according to current cultural principles promotes the importance of the active role of the oncologic patient in the treatment process and management of the ever-changing life needs [1]. It is well known that oncology disease generates a crisis in which patients need to develop strategies and ways to cope and resolve difficulties to regain a balanced condition and learn to take care of themselves. In this condition, patients are frequently willing to establish relationships with healthcare personnel that, through meaningful exchanges, facilitate cooperative management of difficulties and prepare them to acquire different behavioral strategies [2]. In this phase of cultural and care change involving the active participation of patients in the treatment process [1], these patients have become increasingly aware of their own needs and demand to receive information and emotional support through effective communication [3].

In this sense, communication with the healthcare team is a crucial factor in enabling the oncologic patient to define, process, and manage their condition [4] as well as to identify and meet their psychosocial needs [5]. Appropriate communication with healthcare professionals reduces distress and improves pain control, quality of life, and adherence to treatment regimens [6]. Conversely, ineffective communication causes confusion and distress [7].

In this process of learning and support, the central figure among healthcare professionals is the nurse, who has the most frequent and continuous relationship with patients and their families [8]. In addition, the nurse moves simultaneously on different levels, also playing the role of “intermediary” between the patient, family members, and physicians [9]. Nurse–patient communication commonly occurs in an integrated manner with the daily activities of the hospital stay and during the performance of nursing tasks. Communication in this area does not simply refer to exchanging information to achieve better care outcomes but also to moments of emotional exchange [3]. It also offers the possibility of identifying the needs of the patient [10] to facilitate recovery and build realistic and positive goals [11].

The strategies employed by nurses are not always aligned with the inpatient experiences of patients and their families but may reflect the individual’s subjectivity and work organization [12]. In some situations, patients and family members report that they do not feel adequately listened to by healthcare providers and complain of communication difficulties during hospitalization [13].

Communication may, in fact, be affected and hindered by several factors identified as possible barriers that concern both subjects: patients may be reluctant to share their concerns or emotions related to the disease; they may be poorly aware of their condition; they may have a perception of the emotions or manners expressed by the nurses which influence the relationship and the ability to express sensations or needs [14]. Nurses may be unable to pick up signals from patients or manage their emotions and may have a task completion-centered focus or time constraints [15].

From the patient’s perspective, participation, content choice, and communication style are influenced by how nurses handle interactions, e.g., the use of professional terminology, emotional expression, understanding or agreement, and small talk [16]. Patients often take the initiative to talk about emotional concerns [17] and sometimes employ nonverbal signals [18] and strategies, such as humor, to express difficulties or concerns [8].

To understand oncologic patients’ perceptions of the communication process with nurses, a useful approach is to investigate the experience of care reported directly by patients to better interpret and guide improvements in the quality of care in healthcare facilities. Patients’ reported experience is considered an increasingly reliable perspective for understanding their needs and topics of interest, as well as ways to facilitate interaction. Few studies investigate the nurse-patient communication process in oncology within an inpatient department. The existing literature mainly focuses on communication in the terminal stage of the disease or on outpatient oncology patients [19,20].

Given these premises, this study aims to explore which communication methods, as perceived by patients, best meet their needs. Specifically, we want to delve into how certain factors already identified in the literature as determinants in the communication process (such as communication style, nurses’ attitudes, and work organization) shape the experience of oncology patients in the specific setting of hospital admission.

## 2. Materials and Methods

### 2.1. Design

A descriptive, inductive qualitative study was conducted and structured according to the criteria for reporting qualitative research (Consolidated criteria for reporting qualitative research, COREQ) [21,22].

### 2.2. Theoretical Structure

The perception of communication quality in oncology has been extensively studied using both quantitative and qualitative methods. However, for this study, we exclusively employed a qualitative approach. This method allows for access to content and meanings specific to the cultural context in which the investigation takes place, providing in-depth and nuanced insights that are critical for understanding patient experiences. Qualitative studies are particularly important for capturing the nuanced experiences and perspectives of participants, which are often missed by quantitative methods. They allow for a detailed exploration of complex phenomena that may not be easily quantified, thus providing rich, in-depth insights that are essential for understanding patient-nurse communication in oncology settings. This approach aligns with the growing recognition of the importance of qualitative research in healthcare settings [23].

### 2.3. Study Setting and Sample

The population consisted of oncologic patients hospitalized at the Medical Oncology Units of the Regina Elena National Cancer Institute in Rome (IRE). Sampling was purposive [24] to encourage a personal experience of the phenomenon under study and was defined by the principle of data saturation. Data saturation, as proposed by Guest, Namey, and Chen [25], emphasizes the importance of reaching a point where no new information or significant themes emerge from the collected data [25]. Inclusion criteria were the following: age 18 or older, a histologically established cancer diagnosis regardless of disease stage, an inpatient stay in the medical oncology unit of at least 3 days, and willingness to participate in interviews and the study (written informed consent). Patients with cognitive impairment or pathological conditions that might hinder active participation in the study were excluded. To identify these patients, medical records were reviewed, and consultations with attending physicians were conducted to ensure that potential participants met the inclusion criteria.

### 2.4. Instrument

The study used a semi-structured interview, guided by a series of questions based on the researchers’ experiences and the relevant literature. It encouraged participants to share their experiences, with a final question for summarizing suggestions to provide (Table 1).

A socio-demographic and clinical data collection form supplemented the interview.

### 2.5. Data Collection

The interview was conducted from October 2023 to January 2024 within the Medical Oncology Units of the Regina Elena National Cancer Institute in Rome (IRE), operating units of the IRE in a room that allowed for uninterrupted administration and facilitated participants’ comfort and privacy. Two psycho-oncologists conducted the interviews. The interviewers enrolled the patients and explained the survey objectives and data-collection procedures. The enrolled patients were asked to read and sign the informed consent form carefully, thus promoting an informed choice to participate. The questions were tested on the first two participants and did not require any modifications, as they were clear and allowed participants to effectively share their experiences. Data were collected until no new information was emerging (data saturation). The interviews were individual/one-on-one and audio-recorded, to be later transcribed according to the “smooth verbatim transcription” method [26], i.e., reporting the patient’s interview word-for-word and adjusting only the dialectal expressions in Italian. The data that support the findings of this study are openly available at https://gbox.garr.it/garrbox/s/XJXbVnguTXxzLEO (accessed on 18 November 2024).

### 2.6. Data Analysis

Data were analyzed using the Framework Analysis methodology [27]. This type of analysis allows the researcher to explore the data in depth while maintaining effective and transparent control, which improves the rigor of the analytical processes and the credibility of the results. As described by [28], Framework Analysis is an analytical process involving several distinct, highly interconnected steps [29]. The five basic stages outlined are familiarization, identifying a thematic framework, indexing, charting, mapping, and interpretation. Two researchers listened to and transcribed the interviews. Researchers read each interview separately several times, noting key ideas and recurring themes (familiarization). Key ideas and recurring themes were analyzed against the study purpose and interview guide questions (identifying a thematic framework). To ensure accuracy and enable deeper exploration of this trend and subsequent retrieval, the data were labeled into categories (indexing). A summary of the categories under each thematic heading was produced (charting). Graphics were used to map connections between themes and categories and define the phenomenon’s nature (mapping and interpretation) (Table 2). Analyses were performed using NVIVO v. 12 software. To maintain rigor, the study followed the reliability criteria of Guba and Lincoln [28].

### 2.7. Ethical Considerations

The study was conducted by the principles of Good Clinical Practice and in compliance with the requirements of regulatory authorities and key European and national regulations. The study was approved by the Lazio District 5 Territorial Ethics Committee—Verbal Extract no. 3 of 5 September 2023—Trial Register Experiments No. 34/IRE/23.

## 3. Results

### 3.1. Participant Characteristics

The sample consisted of 20 patients. The patients had a mean age of 61.40 years (range: 26–79 years) and were predominantly female (No. 12; 60.0%), diagnosed with lung cancer (No. 11; 55.0%), hospitalized for chemotherapy or radiotherapy (No. 9; 45.0%) (Table 3). Interviews lasted an average of 23 min (range: 15–30 min).

### 3.2. Fundamental Themes and Subthemes

The themes that emerged from the analysis are presented and illustrated below with significant quotes from the participants.

#### 3.2.1. Positive Communication as an Element of Care

##### Communication Is Positive and Satisfactory

Patients reported an overall positive, unhindered, and satisfying communication experience.
*“I am satisfied with how I got along with them.”**(I1)*
*“…I have no complaints about that. I have a very positive experience.”**(I6)*
*“They are immediately well disposed. I really feel comfortable, and I get along well with everyone both day and night—because they take turns day and night.”**(I9)*
*“The truth is that I have never had any problems communicating with them. I never had any obstacles.”**(I13)*
*“I honestly did not find any problems.”**(I17)*

##### Communication as an Element of Care

Patients reported a communication experience as an element of care.
*“It is the kind of communication that heals because you do not feel discomfort and, therefore, you do not say, I do not want to tell them something because I do not feel like it, because I do not like it… you are inclined to communicate willingly with these people.”**(I5)*
*“It is okay because I feel assisted and cared for.”**(I8)*
*“I honestly realized, and this is something I often discuss with the other people here in the room with me as well, that compared with many hospitals, the communication here is very good; it is an element of caring for the person.”**(I9)*
*“The truth is that I get along very well with all the nurses. They are all very good. When you call them, they come immediately and try to comfort you when needed. They take care of you in this way…”**(I13)*

#### 3.2.2. Factors Fostering Communication

##### Informal, Humorous Communication Style, Friendly and Family Atmosphere

Patients reported that the nurses’ friendly and humorous style is one of the facilitating factors in communication and plays an important role in forming relationships that patients refer to as “friendly and familiar.” When it takes on characteristics of confidentiality and intimacy, communication breaks the patients’ isolation and makes them feel part of a relationship and context.
*“Like at home, like one of the family.”**(I2)*
*“I do not want to sound exaggerated, but in my opinion, they are very good. They are nice, they laugh, they joke, we make jokes… we do not look like the patient and the nurse, we almost look like family.”**(I11)*
*“I trust them with my stuff, and they also trust me; I talk about private things, of course, as far as you can go. You get into a closeness that breaks that wall… that detachment… and once you take that away, it is done.”**(I12)*
*“Everyone is nice and ready to chat. They are all young guys, always with a smiling attitude… you do not feel like you are in a hospital, you feel ’like family’ and that is a very important thing.”**(I16)*

##### Simple Interaction, Based on Direct Dialogue and Mediated by Clear Information to Support Choices

Patients reported that communication is facilitated by the simple interaction style, direct dialogue, and the nurses’ ability to provide clear and comprehensive information about their health conditions. The nurses’ expertise in providing accurate information supports patients in their ability to participate in decision-making about their health.
*“I feel that communication between patient and nurse is quite easy; I can dialogue with nurses quite well.”**(I7)*
*“Good communication is when you have a direct dialogue.”**(I17)*
*“…asked for information about anything, it was given to me and very clearly.”**(I19)*
*“Real information, which also supports you at the moment of choice**”**(I19)*

##### Support and Comfort

Patients reported that nurses respond to their requests for help by providing emotional support through nonverbal forms, such as thoughtfulness and attention, and verbal comfort. This modality facilitates patients’ coping with and acceptance of oncology disease.
*“This communication between patients and nurses is a beautiful thing; it helps you deal with… the ’monster’—that is what I call it—that has entered us and that, thanks to God, doctors and nurses can make it disappear. There is no definition… what nurses create with the patient is simply beautiful. It is a type of communication that helps so much and should be in all hospitals, especially for people like me who are very emotional and anxious.”**(I9)*
*“More than this, what else can there be? They stand by us. If we call them, they come immediately and give us a word of comfort. This is a beautiful thing. There are no other things that could be added.”**(I9)*
*“There are few of them—despite that, they always try to give you comfort.”**(I13)*
*“Even though the guys have so much to do when help is needed, they always have a way to do it, and they come right away…”**(I14)*

##### Helpfulness, Attentiveness, and Thoughtfulness

Patients reported that the nurses’ attitude, i.e., helpfulness, attentiveness, and thoughtfulness, facilitates communication; patients feel somewhat free to engage in interactions or ask for help because of the nurses’ disposition.
*“I found all the nurses to be helpful, very accepting, pleasant; so, we had calm, peaceful, and constructive communications.”**(I1)*
*“They are always available about everything and more, you call them, and they… you know, sometimes there can also be the moment when you are a little pain in the neck, but I have never seen these guys angry.”**(I2)*
*“We benefit from their willingness. It is an excellence that this institution holds.”**(I3)*
*“They are always ready for whatever someone needs; they are always ready to explain if someone asks for explanations; they are always smiling; if someone calls them, they are always available to do what is requested.”**(I5)*
*“I am amazed, and they are all good here. They are nice, and even when they come to take your blood samples, they are very gentle and very attentive. They always tell you ‘Thank you’… even in their manners, they are lovely.”**(I16)*

##### Kindness, Humanity, and Empathy

Patients reported that nurses could make them feel understood in their sick condition and simultaneously seen as people through kindness, empathy, and humanity.
*“The nurses are kind; it does not even feel like being in the hospital. It feels like being at home; ‘dear’ over here and ‘Giuliana’ over there are nice things, and a sick person would like to hear just that.”**(I2)*
*“They are very good, and besides patience, they have humanity more than anything else.”**(I2)*
*“The empathy, the grace, the cuddling… the nonsense that is said to you and that comforts you.”**(I8)*
*“They should all hire nurses like the ones inside the Oncology Department of the IFO because they are humane people.”**(I9)*
*“I had no idea there could be such young and nice people. They put you at ease—because sometimes there is also embarrassment in doing some things… and for a moment, you forget you have a woman in front of you. There are tons of kindness here…”**(I12)*
*“Even though it has been a few days, you immediately create that empathy that is important for an oncologic patient like us. Everyone here is like that…”**(I16)*

##### Listening, Reassurance, and the Ability to Instill Hope While Respecting a Sense of Reality

Patients described their relationship with nurses as reassuring because they can rely on them to resolve distress and trust that they will receive the most appropriate care. They cited the crucial ability to realistically instill hope and confidence in times of crisis.
*“Most of the time, I feel understood. For every little thing, they help you; for example, as soon as I say I feel pain, they immediately find the solution and do everything to make me feel good and comfortable.”**(I7)*
*“Unfortunately, here I lost a lady who had undergone surgery with me in orthopedics. I also found her again here in the ward, and I tell you, I was really sad because I saw this person fade away. The nurses were there for me, they talked to me, and they tried to reassure me by saying, ’Do not worry, it is not all like that, do not worry’, and that is something that helped me a lot, a lot, a lot.”**(I9)*
*“They listen when you talk to them; they try to find a solution right away even when you are in pain. They always try to listen to you and solve the problem.”**(I13)*
*“They make you understand that your body will change again… and you will regain what you are now losing. That way, they can give you hope, but it is not fake—that is, they do not tell you unrealistic things like ‘from tomorrow you will come back as if you were 25 years old’. They give you hope concretely, not illusory.”**(I19)*

##### Dynamism, Commitment, and Spirit of Sacrifice

Patients perceive the degree of commitment, spirit of sacrifice, and passion nurses have for their work. These elements reassure patients and reinforce their confidence.
*“Even though the guys have a lot to do when help is needed, they always find a way to do it, and they come right away…”**(I1)*
*“They run. If we want to use a term from Rome, they’ skid.’ They are dynamic and good. They are very active; they do their job with much dynamism. I have been to other hospitals, and it is not like that. I am not naming any names, but the nurses are running here. I have a beautiful relationship with them. They are friendly, kind, and pleasant.”**(I11)*
*“They have passion, and it shows; it transpires. In my opinion, in other hospitals, there are no such nurses. I think they have characteristics that are appropriate for this place. They do a job of professional sacrifice.”**(I12)*

#### 3.2.3. Factors Hindering Communication

##### Lack of Communication Between Nurse and Physician

Patients identified the lack of communication between doctors and nurses as a major obstacle to their care. From the patient’s perspective, the nurse should be more involved in the decision-making and therapeutic process to communicate more effectively with the patient.
*“Having more communication between nurse and doctor… so the nurse could immediately have more options to take action. We as patients communicate much more with nurses than with doctors, whom we only see once a day… so if there was more rapport between them, certainly nurses could have more and faster freedom of action.”**(I15)*

##### Lack of Gratitude, Understanding, Hostility, and Inappropriate Patient Demands

Patients showed a tendency to self-criticize their behavior during hospitalization and reported that some of their attitudes (hostility, lack of gratitude, lack of understanding, inappropriate requests) may be factors hindering communication.
*“They are helpful in everything and do not deny you anything. Of course, if you ask for the moon or the impossible, they cordially say no or ’We will see when the doctor comes’.”**(I10)*
*“Even the sick person must try to have some understanding and respect for the work that they are doing… Always have respect for the work that is being done. They certainly feel so much respect for our disease. A lot.”**(I14)*
*“An obstacle could be perhaps being cranky… maybe a misrepresented word could be hurtful…”**(I18)*
*“I think that if you are hostile, the poor nurse tries tries tries tries but at some point, it is obvious that they will give up if you do not give them any chance.”**(I19)*
*“…for example, if you behave in a certain way, for example, never saying ’Thank you’ when they are doing a job that is important.”**(I20)*

##### Lack of Listening and Understaffing

Patients reported that sometimes experience, in the sense of professional background and role practice, can be an obstacle to truly and authentically listening to each other. At the same time, a staff shortage and lack of time are additional hindering factors in fully accommodating patients’ needs.
*“When they do not listen. When they do not listen to me, I get angry… because I think, ’But if I am saying that…’ […] Not listening to the patient bothers me, which never happened; however, over time, it might happen.”**(I6)*
*“Maybe adding a few more nurses, so they can stop for an extra 2 min and have a chat that, you know, every once in a while, can be helpful to the oncologic patient. I am very honest about that. They are few in the shift… so, if they have to stop for 2 min because they find you in tears because of a personal matter, of course you know it happens once in a while, they cannot do it for too long, because there are so few of them –although, despite that, they always try to comfort you.”**(I13)*
*“Maybe it is sometimes appropriate to put experience aside and listen a little more to the patient. Even if something is not serious but still worries, it is important to listen to it anyway because that means making the patient feel comfortable.”**(I15)*

##### Lack of Welcoming, Attentiveness, Understanding; Lack of Privacy

Patients reported how the lack of intimacy and privacy can be a hindrance for those who feel particularly reserved or demure. They also emphasized how paying attention to even seemingly insignificant aspects could facilitate communication. Finally, they emphasized the individual predisposition necessary for nursing practice.
*“Difficulty of the nurse to interact and to be welcoming with the patient to improve communication, perhaps we should not say to dislodge some people, but at least place suitable people inside the special wards, who can interact well with the patient also through their way of doing and presenting themselves.”**(I5)*
*“It helps a lot to have nurses who understand you, who are very kind to you and who are not… the opposite of being kind.”**(I7)*
*“Perhaps the lack of privacy might hinder communication, although it is not in my case… Intimacy, stuff like that… the point I am making is that I sometimes feel a bit modest…”**(I11)*
*“I know it is difficult; however, in my opinion, it would still be appropriate not to neglect anything… to be attentive… not even the things that seem insignificant because they are important to us.”**(I15)*
*“…I do not like to be pampered; clingy things bother me. But all in all, … maybe yes… I would like, for example, ‘Hi, what are you doing? How are you feeling?’; as I told you before, it is not that I care about these things; however, if they asked me that, it would make me happy”**(I18).*

## 4. Discussion

The study aimed to collect patients’ reported experiences during their hospital stay in two medical oncology wards regarding communication with nursing staff and to explore the factors perceived as facilitating or hindering it.

The results showed that patients generally perceive satisfaction with and appreciation for the communication skills of nurses. Nurses’ communication is found to be appropriate for the oncology care setting. The patients emphasized the difference between the experience of being in an oncology hospital, compared to the previous experience of being in a general one. The results of our study are in disagreement with findings from other studies stating that the psychosocial needs of oncologic patients are often disregarded. That time devoted to communication as an element of care is scarce compared with perceived needs. Differences may relate to the problem of understaffing, the physical environment of the ward, and social factors whereby, in some cultural contexts, patients are discouraged from expressing their needs [29].

The nurses’ interaction style, even before contents or concrete actions, was an effective source of emotional support. Many patients appreciated nurses’ ability to be friendly, talkative, smiling, and funny. Friendly and familiar communication with nurses positively impacts patients’ perceptions of the service and their well-being during hospitalization. It is possible that, in some ways, this style of interaction performs a “cognitive restructuring” function for the patient as the message conveyed is that even the most difficult situations can be handled by relying on “light and positive” relational dimensions. According to Vagnoli L [30], the playful communication style in the nurse-patient relationship helps establish trust, relieves anxiety and tension, and conveys unexpressed emotional messages. In addition, the friendly and familiar atmosphere reported by patients allows the establishment of a bond that helps the patient overcome the feeling of vulnerability resulting from having to rely on an unknown person [8]. The patients themselves actively seek to establish a closeness that makes them feel not just numbers but recognized in their uniqueness, as happens in a family setting [8].

Clear and effective communication with nurses is another element that patients mentioned as facilitating and fundamental in ensuring adequate support during the disease journey. In the interviews, patients narrated how important it was to receive clear and direct information and how this helped them cope with the difficulties associated with their disease. The information was consistent, i.e., responsive to the needs expressed while respecting their learning time. Patients were awe-struck by a nurse taking the time to give information, which gratified them and made them feel included despite the heavy workload and staff shortage. Less technical language focused on patient understanding is recognized as a powerful tool for providing better clinical care.

Providing adequate patient information is a crucial aspect of nursing care with the goal of caretaking. Literature studies confirm how clear and effective communication between patients and healthcare providers can increase understanding of medical information, improve adherence to prescribed therapies [31], improve symptom management, and reduce anxiety and depression in patients with serious or chronic diseases.

Another element that emerged as facilitating communication relates to nurses’ attitudes: helpfulness, attentiveness, and thoughtfulness promote interactions and produce well-being in patients. These are modalities referred to as “caring,” i.e., the intentional and conscious action of conveying physical care and emotional concern toward the other. These behaviors carry meaningful messages to the other person and do not require additional time or resources; patients did not emphasize practices, such as long dialogues or lengthy interactions, but rather modes that provide presence and a sense of security.

Kindness, humanity, and empathy were also reported as elements of caring that positively influence patient experience and improve the quality of care and clinical outcomes. Empathic communication is associated with greater emotional involvement of patients, greater satisfaction with the experience of care, and adherence to treatments [32].

In addition, a person-centered approach can increase patient trust in nurses, reduce pain and anxiety, and help improve patient clinical outcomes and quality of life [33].

The interviews also revealed some barriers to communication in the oncology setting that negatively affect the patient-nurse relationship and, thus, the care experience. Some patients reported that a major obstacle was the lack of communication and collaboration among healthcare team members, which could instead enable an even more efficient response to patient needs. Literature studies report that the absence of a cohesive and collaborative multidisciplinary team can negatively impact care outcomes [34]; nurses frequently face the challenge of gathering and interpreting the various aspects of the patient’s treatment plan, including information from clinical visits and different physicians [35].

Some patients self-critically stated that their attitudes may influence and hinder good communication, such as a lack of gratitude, hostility, or inappropriate requests. A study reports precisely how misunderstandings can create tension in the relationship and hinder effective communication [36].

Patients reported that a further obstacle to authentic and equal interaction is communication centered on “exercising a role” and “professional baggage” rather than on the person and the present experience; in fact, the literature confirms that, in some situations, nurses, despite having the necessary communication competences, tend to perform in a task-centered manner as a protective mechanism against the emotional or advocacy aspects of their profession [5].

Patients highlight that the shortage of nursing staff and the limited time nurses can dedicate to them is a significant barrier to communication. This finding is consistent with the previous literature [37].

Finally, patients stressed the importance of the elements related to caring and emphasized the individual predisposition, probably intended as motivation and interest in the practice. From their point of view, the lack of hospitality, attention, kindness, and privacy can significantly compromise the communication process. The experience reported by patients is consistent with nurses’ perception of their role: “adaptability” is understood as the ability to fluidly connect knowledge and experience, leadership, motivation, patient-centered care, organization, and culture are some of the most important factors [38].

Interestingly, most of the factors patients reported as facilitating communication refer to spontaneous resources, which do not only necessarily come through training but are developed through historical, handed-down experience and shared group practice. In this sense, it appears that the nursing group defines and differs from other healthcare professionals based on what it understands and consciously recognizes as specific characteristics of the role: caring is part of a culture that this specific professional group acquired and established. Therefore, as for the organization and management within Nursing Institutions, this study underscores how important it is to create or reinforce, where it already exists, professional socialization to sustain a group culture based on relationship effectiveness, including its more spontaneous forms, accompanied by role-specific competencies. At the university level, training related to patient-centered communication could focus on showing that this type of communication is not time-consuming and is among the prerequisites that patients recognize as fundamental to quality care facilitating their course of treatment.

In summary, it would be important for both educational institutions and healthcare organizations to consider focusing on the empowerment of nurses’ relational skills and the importance of an organizational system that promotes their expression.

### 4.1. Work Strengths and Limitations

Qualitative methodology encompasses strengths and weaknesses. Patients reported experience is a perspective considered increasingly reliable for understanding their needs, the content they prefer to explore in-depth, and the ways that facilitate interaction. The limitations of this methodology may be the extreme individuality and subjectivity with which data are collected. For this reason, it is often difficult to replicate the inferential process that leads to the generalization of the data to the general population. Additionally, the qualitative approach may struggle with researcher bias, as the researcher’s perspective can influence data interpretation. Moreover, the lack of standardization in data collection and analysis methods can lead to variability in results. The study also investigates the patients’ experience in a specific cultural context and setting, the city of Rome, making it difficult to generalize the data to clinical and cultural contexts that follow different behavioral and relational rules.

### 4.2. Recommendations for Further Research

Building on the results of our study, it would be useful to explore the ways and strategies of acquiring communication competencies in the group of nurses by delving into what is learned during the educational process and what pertains to the culture and history that groups can generate during their professional experience.

### 4.3. Implications for Practice

The study results will guide nurses toward understanding an effective method of communication with the oncologic patient and, thus, toward improving standards of care and assistance. Furthermore, this can lead to the improvement of the advanced competencies of nurses.

## 5. Conclusions

This study highlights that oncology patients generally perceive their communication with nursing staff as satisfactory, primarily due to the positive, caring communication style of nurses. Patients emphasized that factors, such as a familiar communication style, humor, clear information, and emotional support, significantly enhance their care experience. Conversely, barriers such as lack of communication between nurses and physicians, understaffing, and insufficient time for interactions hinder effective communication. To improve patient well-being and care outcomes in oncology settings, it is crucial to enhance communication skills between nurses and physicians and foster a collaborative environment within healthcare teams. The results indicate that communication between nursing staff and patients is satisfactory. However, what is lacking is effective communication between nurses and physicians. Therefore, we recommend that healthcare institutions prioritize training in advanced communication competencies and encourage a culture of empathy and support to better meet the psychosocial needs of cancer patients.

## Figures and Tables

**Table 1 healthcare-13-00050-t001:** Semi-structured interview questions.

Could you describe your experience of communicating with the nursing staff on the ward? If positive—What were the factors promoting communication? What could be the factors hindering communication? If negative—What were the factors hindering communication? What could be the factors promoting communication? Do you have any suggestions for improving communication between patients and nursing staff?

**Table 2 healthcare-13-00050-t002:** Example of the coding process in inductive analysis.

No. Interview	Codes	Categories	Theme
1	I was fine	Communication is satisfactory	Positive communication as an element of care
5	I could only speak highly of the people here		
6	I have a very positive experience		
1	I am satisfied with how comfortable I was		
3	For me, the service is excellent		
13	I have never had any problems communicating with them		
17	I have not found any issues		
9	Communication here is very good; it is an element of care	Communication as an element of care	
13	They take care of you in this way		
8	It is good because I feel assisted and cared for		
5	It is the kind of caring communication		

This matrix was used for each interview. This excerpt refers to the first theme that emerged.

**Table 3 healthcare-13-00050-t003:** Socio-demographic and clinical data.

Age	Average: 61.4 years	Range: 26–79 years
	N.	%
Gender:		
Female	12	60.0%
Male	8	40.0%
Marital status:		
Single	2	10.0%
Married/Cohabiting	10	50.0%
Divorced/Separated	5	25.0%
Widowed	3	15.0%
Education level:		
Elementary school	2	10.0%
Middle school	4	20.0%
High school	10	50.0%
Bachelor’s degree	4	20.0%
Pathology:		
Cervical cancer	1	5.0%
Colon cancer	1	5.0%
Tongue cancer	1	5.0%
Lung cancer	11	55.0%
Stomach cancer	1	5.0%
Osteosarcoma	1	5.0%
Sarcoma	3	15.0%
Thymoma	1	5.0%
Current treatment:		
Chemotherapy	15	75.0%
Radiotherapy	1	5.0%
None	4	20.0%

## Data Availability

The data that support the findings of this study are openly available at https://gbox.garr.it/garrbox/s/XJXbVnguTXxzLEO (accessed on 18 November 2024).

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
