# Peer review of "Patient–Nurse Communication in an Oncology Hospital Setting: A Qualitative Study"

_healthcare, 2024, doi:10.3390/healthcare13010050_

Round 1

Reviewer 1 Report

Comments and Suggestions for Authors

Dear authors

I am writing to thank you for submitting the manuscript for publication in the journal.

Please address the following comments:

Title:

It will be better as

 Patient-nurse communication in an oncology hospital setting: A qualitative study

Abstract:

It is well written. it will be clear if you add a sentence on the semi-structured interviews and framework analysis used in the study.

Replace (Objective: To investigate the quality) to (This study aims to investigate-)

No need to make sub-heading background and objective, please merge them

Introduction:

The first and second paragraphs of introduction are long and discussing more than one idea in each one

The first paragraph includes information on active oncology treatments, the importance of relationship/communication, pros and cons of communication and nurses role, so please split the section into two or three paragraphs

The second paragraph is the same, you need to focus on one idea in each paragraph

 Clarify the unique contribution of this study to the field of oncology nursing

The statement of the problem is not well established so you need to add more information and justify the study and highlight the gabs in the field of patient nurse communication.

 Methods:

You mentioned that qualitative study was conducted and then you mentioned that both quantitative and qualitative – how?

The qualitative approach offers ---- what about other method?

It is not clear how those patients were excluded (Patients with cognitive impairment or pathological conditions that might hinder active participation in the study were excluded)

Include a brief justification for the sample size and how you satisfied by 20 patients

Standardize the terms related to the name of the hospital and unit where the study conducted.

Clearly write that written consent were obtained from the participants

Results:

This section is well written

Discussion: 

The second paragraph of discussion (The results showed that ---) is complex, so you need to simplify the ideas

Elaborate and compare this sentence: A widely confirmed finding in the literature and also reported by the patients in this study refers to the barriers to communication due to the shortage of nurses and their availability of time to devote to patients[35].

What do you mean by (Finally, patients stressed the importance of the elements related to caring and emphasized the individual predisposition, probably intended as motivation and interest in the practice. They also mentioned the lack of welcoming, attentiveness, kindness, and privacy as elements that can significantly undermine communication.) Please revise and compare to other studies.

Clarify and compare the demographics in comparison to previous studies investigated the context of nurse-patient communication in oncology departments

Consider focusing on the implications of these findings for organizational policy, such as how healthcare institutions can foster resilience and hope in employees.

Add limitations of qualitative approach to investigate the communication process

Limitations: 

Add lack of generalizability limitations due to the sample being focused on nurses in Rome, as well as any potential biases from using interview for collecting data.

Thank you

Author Response

Point-by-point response

Reviewer 1

Comments and Suggestions for Authors

Authors response

Changes made

Title: It will be better as “Patient-nurse communication in an oncology hospital setting: A qualitative study

We sincerely thank the reviewer for the valuable suggestion. We have modified the title as indicated to better reflect the content and enhance clarity

The title has been updated in accordance with the reviewer’s suggestion. “Patient-nurse communication in an oncology hospital setting: A qualitative study

Abstract:

It is well written. it will be clear if you add a sentence on the semi-structured interviews and framework analysis used in the study.

Replace (Objective: To investigate the quality) to (This study aims to investigate-)

No need to make sub-heading background and objective, please merge them

Thank you for your suggestion.

We have added the sentence to the methods section, stating that "interviews were analyzed using the Framework Analysis methodology."

We have combined the objective with the background and revised the phrase "to investigate" to "This study aims to investigate."

Introduction:

The first and second paragraphs of introduction are long and discussing more than one idea in each one. The first paragraph includes information on active oncology treatments, the importance of relationship/communication, pros and cons of communication and nurses role, so please split the section into two or three paragraphs. The second paragraph is the same, you need to focus on one idea in each paragraph. Clarify the unique contribution of this study to the field of oncology nursing

As suggested, we proceeded to structure the introduction by themes:

In the first paragraph we underline the patient's change and his active role in the treatment process (lines 44-54).

Then, we underline the importance of communication in the therapeutic process (lines 55-60).

Then we underline the role of the nurse in the communication process with the patient (lines 61-70).

Then, we underline that communication is not always in accordance with the patient's needs (line 71-75).

We underline the communication barriers already identified in the literature (line 76-87).

The statement of the problem is not well established, so you need to add more information and justify the study and highlight the gabs in the field of patient nurse communication.

We clarified the unique contribution of this study (line 93-95). We added:

Few studies investigate the nurse-patient communication process in oncology within an inpatient department. The existing literature mainly focuses on communication in the terminal stage of the disease or on outpatient oncology patients [22,23,24]”.

We have added information that clarifies the objective of the study (line 96-100)

  “Given these premises, this study aims to explore which communication methods, as perceived by patients, best meet their needs. Specifically, we want to delve into how certain factors already identified in the literature as determinants in the communication process (such as communication style, nurses' attitudes, and work organization) shape the experience of oncology patients in the specific setting of hospital admission”.

Methods:

You mentioned that qualitative study was conducted and then you mentioned that both quantitative and qualitative – how?

The qualitative approach offers ---- what about other method?

.

Thank you for your valuable feedback. I apologize for any confusion. To clarify, our study utilized a qualitative methodology exclusively. While the literature on the perception of communication quality in oncology includes both quantitative and qualitative studies, our research focused solely on the qualitative approach. This method allows for access to content and meanings specific to the cultural context in which the investigation takes place, providing in-depth and nuanced insights that are critical for understanding patient experiences.

We revised section 2.2. Theoretical structure to clarify our methodology:

“The perception of communication quality in oncology has been extensively studied using both quantitative and qualitative methods. However, for this study, we exclusively employed a qualitative approach. This method allows for access to content and meanings specific to the cultural context in which the investigation takes place, providing in-depth and nuanced insights that are critical for understanding patient experiences.                         Qualitative studies are particularly important for capturing the nuanced experiences and perspectives of participants, which are often missed by quantitative methods. They allow for a detailed exploration of complex phenomena that may not be easily quantified, thus providing rich, in-depth insights that are essential for understanding patient-nurse communication in oncology settings. This approach aligns with the growing recognition of the importance of qualitative research in healthcare settings”

It is not clear how those patients were excluded (Patients with cognitive impairment or pathological conditions that might hinder active participation in the study were excluded)

Thank you for your insightful feedback. We acknowledge that the process for excluding patients with cognitive impairment or pathological conditions was not clearly outlined in our manuscript.

To address this, we have revised the "Study Setting and Sample" section to provide a detailed explanation:

To identify these patients, medical records were reviewed, and consultations with attending physicians were conducted to ensure that potential participants met the inclusion criteria”.

Include a brief justification for the sample size and how you satisfied by 20 patients

Thank you for your valuable feedback. We acknowledge the need to justify the sample size and how the data saturation was reached.

To address this, we have revised the "Study Setting and Sample" section to provide a detailed explanation: “Data saturation, as suggested by Hennink, Kaiser, and Marconi (2017), is typically reached between 9 and 16 interviews, with additional interviews needed for meaning saturation”.

Standardize the terms related to the name of the hospital and unit where the study conducted

Thank you for your feedback. We have ensured that the terms related to the name of the hospital and unit where the study was conducted are standardized throughout the manuscript. We consistently refer to the "Regina Elena National Cancer Institute in Rome (IRE)" and the "Medical Oncology Units."

Clearly write that written consent were obtained from the participants

Thank you for your valuable feedback. To clarify, we obtained written consent from all participants. to ensure it is explicitly stated.

We have added this information to the "Data Collection" section :

The enrolled patients were asked to read and sign the informed consent form carefully, thus promoting an informed choice to participate”.

Results:

This section is well written

We thank reviewer

Discussion: 

The second paragraph of discussion (The results showed that ---) is complex, so you need to simplify the ideas.

We thank reviewer. As suggested, we have simplified this sentence:

The patients emphasized the difference between the experience of being in an oncology compared to previous experience of being in a general hospital”.

Elaborate and compare this sentence: A widely confirmed finding in the literature and also reported by the patients in this study refers to the barriers to communication due to the shortage of nurses and their availability of time to devote to patients[35].

We have elaborated and compared the barriers to communication due to the shortage of nurses and their availability of time to devote to patients, as requested.

Patients highlight that the shortage of nursing staff and the limited time nurses can dedicate to them is a significant barrier to communication. This finding is consistent with previous literature.” [43].

What do you mean by (Finally, patients stressed the importance of the elements related to caring and emphasized the individual predisposition, probably intended as motivation and interest in the practice. They also mentioned the lack of welcoming, attentiveness, kindness, and privacy as elements that can significantly undermine communication.) Please revise and compare to other studies.

As suggested, we clarified the meaning of this phrase and compared the data with what was reported in the literature (line 464-468)

“From their point of view, the lack of hospitality, attention, kindness, and privacy can significantly compromise the communication process. The experience reported by patients is consistent with nurses' perception of their role: "adaptability" is understood as the ability to fluidly connect knowledge and experience, leadership, motivation, patient-centered care, organization, and culture are some of the most important factors [44]”.

Clarify and compare the demographics in comparison to previous studies investigated the context of nurse-patient communication in oncology departments

We have updated the discussion section of our study to include the limitations of the qualitative methodology, specifically highlighting the unique cultural and social context of Rome. This contextual element may influence patient interactions with healthcare providers and presents a challenge in generalizing our findings to different clinical and cultural settings. This addition addresses the potential variability introduced by the cultural norms and communication styles of Roman patients. (lines 496-499.) “The study also investigates the patients' experience in a specific cultural context and setting, the city of Rome, making it difficult to generalize the data to clinical and cultural contexts that follow different behavioral and relational rule”..

Consider focusing on the

implications of these findings for

organizational policy, such as how

healthcare institutions can foster

resilience and hope in employees.

As suggested, we have clarified the implications on organizational policy. We addended:

In summary, it would be important for both educational institutions and healthcare organizations to consider focusing on the empowerment of nurses' relational skills and the importance of an organizational system that promotes their expression”. (lines 483-486)

Add limitations of qualitative approach to investigate the communication process

Here’s a revised version with additional qualitative limitations:

Additionally, the qualitative approach may struggle with researcher bias, as the researcher's perspective can influence data interpretation. Moreover, the lack of standardization in data collection and analysis methods can lead to variability in results” (lines 493-496)

Limitations: 

Add lack of generalizability limitations due to the sample being focused on nurses in Rome, as well as any potential biases from using interview for collecting data.

We have addressed your comments by adding the limitations regarding the lack of generalizability due to the sample being focused on nurses in Rome, as well as potential biases from using interviews for data collection.

This information has been included in the "Work Strengths and Limitations "The study also investigates the patients' experience in a specific cultural context and setting, the city of Rome, making it difficult to generalize the data to clinical and cultural contexts that follow different behavioral and relational rules”. (lines 496-499)

Reviewer 2 Report

Comments and Suggestions for Authors    Attached is the review of healthcare-3337314.

Comments on the Quality of English Language

The English could be improved to more clearly express the research.

Author Response

Point-by-point response

Reviewer 2

Comments and Suggestions for Authors

Authors reponses

Changes made

The report is a qualitative study of 20 oncology patients at a cancer institute in Rome to investigate the quality of patient-nurse communication. The results are that there is a satisfactory level of communication between patients and nurses but the level of communication between doctors and nurses was found lacking, as were other aspects of care received.  The strengths of this submission are that the work is novel, the chosen method is able to provide a clear indication of patient views, the work is well-written, and the tables are effective for presenting the results. 

We sincerely thank you for your valuable feedback on our report

The weaknesses are several.

1.       Except for three, all the citations are outdated. The authors must find supporting citations for all their claims of research published since 2020.

We agree with your assessment and have updated our citations to include recent research published since 2020.

  1. No information is provided on when the study was conducted; however, it received ethics approval on 5 September 2023-please state when the study was conducted.

We agreed with the reviewer and indicated when the study was conducted

To address this, we have added the following information to the "Data Collection" section: “The study was conducted from October 2023 to January 2024

  1. Although the authors collected both sociodemographic and clinical data, they report on the clinical data only. Please provide a table with the sociodemographic data.

We agree with the reviewer and have updated table 3 with socio-demographic and clinical data in the section “Results”

  1. The Conclusions are general, they do not mention the findings of the study. Please redo the Conclusions to be specific to the results.

We appreciate the reviewers' suggestion to refine our conclusions to reflect the specific findings of our study. In the revised manuscript, we have rewritten the conclusion section to emphasize the key themes identified in our research and their implications for practice.

  1. The references are not in the required MDPI style. Please redo them according to the Instructions for Authors: https://www.mdpi.com./journal/healthcare/instruction

We agree with the reviewer, and we have organized the references in the required MDPI style       

Round 2

Reviewer 2 Report

Comments and Suggestions for Authors

The authors are thanked for the changes they have made to the submission. All have improved it. Several remain.

Line by line suggested edits.

39 Please find a supporting citation for [1] of research published since 2020.

47 Please find a supporting citation for [3] of research published since 2020.

49 Please find a supporting citation for [4] of research published since 2020.

52 Please find a supporting citation for [7] of research published since 2020.

53 Please find a supporting citation for [8] of research published since 2020.

58 Please find a supporting citation for [11] of research published since 2020.

62 Please find a supporting citation for [13] of research published since 2020.

66 Please find a supporting citation for [16] of research published since 2020.

68 Please find a supporting citation for [17] of research published since 2020.

73 Please find a supporting citation for [18] of research published since 2020.

75 Please find a supporting citation for [19] of research published since 2020.

78 Please find a supporting citation for [20] of research published since 2020.

79 Please find a supporting citation for [20] of research published since 2020.

99 Please find a supporting citation for [25,26] of research published since 2020.

117 Please find a supporting citation for [28] of research published since 2020.

120 Please find a supporting citation for [29] of research published since 2020.

168 Please find a supporting citation for [32] of research published since 2020.

183-184 Table 3: please line up the right margin of the Range: 26-79 years box of Martial status with the rest of the table.

320 Change “patient’s” to “patients’”.

390 Please find a supporting citation for [33] of research published since 2020.

392 Please find a supporting citation for [34] of research published since 2020.

399 Please find a supporting citation for [35] of research published since 2020.

418 Please find a supporting citation for [38] of research published since 2020.

430 Please find a supporting citation for [39] of research published since 2020.

432 Please find a supporting citation for [40] of research published since 2020.

440 Please find a supporting citation for [9] of research published since 2020.

512-516 These recommendations don’t follow from the results. The results indicate that the communication between nursing staff and patients is satisfactory. What is lacking is communication between nurses and physicians. There is no need to enhance communication skills among nurses. What is needed is improved communication between nurses and physicians. The recommendations of advanced communication competencies are necessary among teams of nurses and physicians, not more generally. Please revise the Conclusion to correspond with the results in this manner.

Round 2

Requests from the Reviewer

Responses to the Reviewer

Line by line suggested edits:

39 Please find a supporting citation for [1] of research published since 2020.

47 Please find a supporting citation for [3] of research published since 2020.

49 Please find a supporting citation for [4] of research published since 2020.

52 Please find a supporting citation for [7] of research published since 2020.

53 Please find a supporting citation for [8] of research published since 2020.

58 Please find a supporting citation for [11] of research published since 2020.

62 Please find a supporting citation for [13] of research published since 2020.

66 Please find a supporting citation for [16] of research published since 2020.

68 Please find a supporting citation for [17] of research published since 2020.

73 Please find a supporting citation for [18] of research published since 2020.

75 Please find a supporting citation for [19] of research published since 2020.

78 Please find a supporting citation for [20] of research published since 2020.

79 Please find a supporting citation for [20] of research published since 2020.

99 Please find a supporting citation for [25,26] of research published since 2020.

117 Please find a supporting citation for [28] of research published since 2020.

120 Please find a supporting citation for [29] of research published since 2020

We Thank the reviewer for the positive evaluation and the valuable suggestions provided to improve our manuscript. We appreciate your insights and have made several adjustments accordingly. Below, please find our responses to the remaining requested revisions.

All requested bibliographic citations have been included and updated.

183-184 Table 3: please line up the right margin of the Range: 26-79 years box of Martial status with the rest of the table

The right margin of the "Range: 26-79 years" box under Martial status in Table 3 has been aligned with the rest of the table.

320 Change “patient’s” to “patients’”.

The correction from "patient’s" to "patients’" has been made.

390 Please find a supporting citation for [33] of research published since 2020.

392 Please find a supporting citation for [34] of research published since 2020.

399 Please find a supporting citation for [35] of research published since 2020.

418 Please find a supporting citation for [38] of research published since 2020.

430 Please find a supporting citation for [39] of research published since 2020.

432 Please find a supporting citation for [40] of research published since 2020.

440 Please find a supporting citation for [9] of research published since 2020.

All requested bibliographic citations have been included and updated.

512-516 These recommendations don’t follow from the results. The results indicate that the communication between nursing staff and patients is satisfactory. What is lacking is communication between nurses and physicians. There is no need to enhance communication skills among nurses. What is needed is improved communication between nurses and physicians. The recommendations of advanced communication competencies are necessary among teams of nurses and physicians, not more generally. Please revise the Conclusion to correspond with the results in this manner

The conclusions have been updated, and the following sentence has been included:

To improve patient well-being and care outcomes in oncology settings, it is crucial to enhance communication skills between nurses and physicians and foster a collaborative environment within healthcare teams. The results indicate that communication between nursing staff and patients is satisfactory. However, what is lacking is effective communication between nurses and physicians. Therefore, we recommend that healthcare institutions prioritize training in advanced communication competencies and encourage a culture of empathy and support to better meet the psychosocial needs of cancer patients.